# Root Transcriptional and Metabolic Dynamics Induced by the Plant Growth Promoting Rhizobacterium (PGPR) *Bacillus subtilis* Mbi600 on Cucumber Plants

**DOI:** 10.3390/plants11091218

**Published:** 2022-04-30

**Authors:** Anastasios Samaras, Nathalie Kamou, Georgios Tzelepis, Katerina Karamanoli, Urania Menkissoglu-Spiroudi, George S. Karaoglanidis

**Affiliations:** 1Laboratory of Plant Pathology, School of Agriculture, Faculty of Agriculture, Forestry and Natural Environment, Aristotle University of Thessaloniki, 54124 Thessaloniki, Greece; gkarao@agro.auth.gr; 2Pesticide Science Laboratory, School of Agriculture, Faculty of Agriculture, Forestry and Natural Environment, Aristotle University of Thessaloniki, 54124 Thessaloniki, Greece; nnkamou@gmail.com (N.K.); rmenkis@agro.auth.gr (U.M.-S.); 3Department of Forest Mycology and Plant Pathology, Swedish University of Agricultural Sciences, Uppsala Biocenter, Box 7026, SE-750 07 Uppsala, Sweden; georgios.tzelepis@slu.se; 4Laboratory of Agricultural Chemistry, School of Agriculture, Faculty of Agriculture, Forestry and Natural Environment, Aristotle University of Thessaloniki, 54124 Thessaloniki, Greece; katkar@agro.auth.gr

**Keywords:** *Bacillus* MBI600, transcriptomic analysis, metabolomic analysis, defense-related proteins, plant growth promotion

## Abstract

*Bacillus subtilis* MBI600 is a commercialized plant growth-promoting bacterial species used as a biocontrol agent in many crops, controlling various plant pathogens via direct or indirect mechanisms. In the present study, a detailed transcriptomic analysis of cucumber roots upon response to the *Bs* MBI600 strain is provided. Differentially expressed genes (DEGs) analysis showed altered gene expression in more than 1000 genes at 24 and 48 h post-application of *Bs* MBI600. *Bs* MBI600 induces genes involved in ISR and SAR signaling. In addition, genes involved in phytohormone production and nutrient availability showed an upregulation pattern, justifying the plant growth promotion. Biocontrol ability of *Bs* MBI600 seems also to be related to the activation of defense-related genes, such as peroxidase, endo-1,3(4)-beta-glucanase, PR-4, and thaumatin-like. Moreover, KEGG enriched results showed that differentially expressed genes were classified into biocontrol-related pathways. To further investigate the plant’s response to the presence of PGPR, a profile of polar metabolites of cucumber treated with *Bs* MBI600 was performed and compared to that of untreated plants. The results of the current study gave insights into the mechanisms deployed by this biocontrol agent to promote plant resistance, helping to understand the molecular interactions in this system.

## 1. Introduction

Cucumber (*Cucumis sativus* L.) is a creeping vine plant in the Cucurbitaceae family, widely cultivated all over the world either in open fields or in greenhouses. Cucumber cultivation is characterized by its high requirements for fertilizer and pesticide applications, since it can be infected by a plethora of foliar and soil-borne fungal pathogens [1,2]. Nowadays, most of the nutrients necessary for plant growth in agricultural production are mainly provided by chemical fertilizers [3]. In addition, the control of plant pathogens is based mostly on chemical fungicide applications, with numerous negative effects on human health and the environment. In an attempt to reduce the side effects of chemical use, application of beneficial microorganisms can be an alternative solution and has gained ground in horticultural production.

The soil environment surrounding the plant roots, the rhizosphere, is the area where plants interact with microorganisms, including plant growth-promoting rhizobacteria (PGPR) [4,5]. PGPR exhibit a beneficial impact on plants by producing enzymes, secondary metabolites, phytohormones, or other compounds, and contribute to plant growth through different mechanisms [6,7]. One of the most dominant practices is phytostimulation through enhancing plant nutrition, the production of phytohormones, or by modulation of hormone homeostasis in plants [4]. The role of PGPR in hormonal plant-signaling pathways is widely known, but more studies are needed to confirm their precise role in this procedure.Previous reports showed that treatment with the Gram-negative bacterial species *Phyllobacterium brassicacearum* STM196 increased lateral roots formation in *Arabidopsis* plants, by triggering changes in IAA distribution and homeostasis independently from IAA bacterial production [8] For instance, *Pseudomonas* strains were reported to be involved in auxin signaling and transport in *Arabidopsis* plants [9]. Most Gram-positive bacteria can produce indole acetic acid (IAA), the most common phytohormone with a potential role in plant-microbe interactions [10].

Even though numerous PGPR strains have been used since the 1990s to control different pathogens on cucumber, knowledge regarding the molecular aspects of these interactions is limited.Furthermore, PGPR have also been shown to induce systemic resistance (ISR) through the activation of different defense-related pathways, connected to salicylic (SA) and jasmonic (JA) acid, or to ethylene (ET) signaling [11,12,13,14]. The first report of ISR induction on cucumber plants treated with PGPR was in 1991, when Wei et al. (1991) screened 94 PGPR strains for induction of cucumber resistance against the anthracnose agent *Colletotrichum orbiculare* [15]. In a limited number of studies, enzymes, such as peroxidases (PO), polyphenol oxidases (PPO), superoxide dismutase (SOD), and catalases, were activated on cucumber roots challenged with *Pseudomonas* spp. or *Bacillus* spp. strains [16,17]. 

*Bacillus subtilis* MBI600 (thereafter *Bs* MBI600) is a biological control agent (BCA) that has been recently commercialized by BASF; however, its genetic characterization and mode of PGP activity need to be elucidated. Recently, we unraveled its taxonomy using whole genome sequencing approaches, and several genes associated with biofilm formation, nutrient uptake, and antibiotic production were identified in its genome [18]. In the same study, we showed that *Bs* MBI 600 was able to colonize cucumber roots and induce increases in shoot and root length, while also able to reduce disease severity caused by *Pythium aphanidermatum* and *Fusarium oxysporum* f.sp. *radicis-cucumerinum* [18]. Moreover, *Bs* MBI 600 was able to induce certain resistant genes in this host [19]. However, molecular mechanisms behind these responses of cucumber plants remain largely unknown. Furthermore, on other hosts, such as tomato, *Bs* MBI600 was found to be efficient in providing resistance to viral infections by *TSWV* or *PVY* through a dose-dependent, synergistic interaction of salicylic acid (SA) signaling and jasmonic acid (JA) signaling pathways [20,21]. Similarly, *Bs* MBI600 was found to be efficient in promoting the growth of tomato plants and controlling *Fusarium* crown and root rot caused by *F. oxysporum* f.sp. *radicis-lycopersici* [22,23].

An investigation of the genetic and molecular mechanisms activated in plants as a response to their exposure to PGPR may help towards to the optimization of their use in modern agriculture. Novel technologies, such as next-generation sequencing (NGS), have recently been established as the key tool for understanding the taxonomic and functional behavior of PGPR. Furthermore, the development of meta “omic” technologies could help to gain deeper knowledge of complex “plant-PGPR.” Recently, RNA sequencing (RNA-seq) has been established as a useful tool for transcriptome analysis in many plant species to identify their responses to PGPR applications and understand the molecular basis of the activated mechanisms [24,25]. Moreover, the combination of metabolite and transcript profiling data offers a holistic approach to the study of the responses of plants inoculated with a PGPR strain. The metabolic composition of specific sampling times can provide a deeper explanation of the phenotype, always in accordance with the RNA-seq results.

Thus, in the present study, we investigated the effects of *Bs* MBI600 on the transcriptome and metabolome levels of cucumber roots. Genes and metabolic compounds with important roles were identified, mainly related to the resistance to pathogens and to plant growth induction, and we provided further insights at the transcriptome level into the observed growth promotion and resistance of cucumber against soilborne pathogens.

## 2. Results

### 2.1. Treatment with B. subtilis MBI600 Increased Plant Growth

To determine whether the root-drenching applications of *Bs* MBI600 induce the growth of cucumber plants, several growth parameters were measured. The results confirmed that *Bs* MBI600 treatment significantly increased (*p* < 0.05) shoot height compared to untreated plants, but for the remaining parameters measured (shoot fresh weight, shoot dry weight, root length, root fresh weight, and root dry weight) no differences (*p* > 0.05) were observed among *Bs* MBI600-treated and untreated plants (Figure 1). Applications of the biological reference treatment *Ba* QST713 resulted in higher shoot height, shoot fresh weight, root length, and root fresh weight compared to the untreated plants (Figure 1).

In addition, two photosynthetic parameters were measured to investigate changes after *Bs* MBI600 application. Treatment of cucumber plants with either *Bs* MBI600 or *Ba* QST713 did not affect the chlorophyll content index (CCI) (*p* > 0.05), as compared to untreated plants. However, net photosynthesis was found to have increased (*p* < 0.05) on *Bs* MBI600-treated plants, while remaining unaffected on *Ba* QST713-treated plants (Appendix A).

### 2.2. Differential Gene Expression in Response to Bs MBI600

To understand more thoroughly the molecular interactions between cucumber and *Bs* MBI600, the transcriptome profile of the plant was investigated at different time points after inoculation with the BCA. Samples from treated plants were compared to those before the application of *Bs* MBI600. Raw data were analyzed as reads counts, normalized, and mapped in the Cucumis sativus genome in 78% coverage. Initially, a volcano plot analysis was conducted to identify the differentially expressed genes (DEGs) between untreated plants (control) and *Bs* MBI600-treated plants, 24 h (MBI24) or 48 h (MBI48) post-application; MBI24 vs. Control, MBI48 vs. Control, and MBI48 vs. MBI24 (Appendix A). A differential expression analysis was performed using edgeR in the following comparison style: MBI24 vs. Control, MBI48 vs. Control, and MBI48 vs. MBI24, and the genes (DEGs) were identified in 1922, 1372, and 2707, respectively (Appendix A). According to our analysis, the comparison between MBI48 vs. MBI24 treatments showed a higher number of DEGs with significant difference (Appendix A). The top 400 DEGs that showed the most significant differences in roots 24 and 48 h post-application are presented in Figure 2.

In the MBI48 vs. MBI24 comparison, the higher number of up-regulated genes (1777) was observed, followed by MBI48 vs. Control (727) and MBI24 vs. Control (692). Interestingly, only few identical DEGs were identified at both time points. In detail, 33 genes were up-regulated, and 63 were down-regulated (Figure 3).

### 2.3. DEGs Involved in PGPR Biocontrol Mechanisms

To identify the potential function of the DEGs, a gene annotation was conducted. The three main gene categories associated with responses to the *Bs* MBI600 treatment included genes involved in signaling, defense against microorganisms, and plant growth (directly or indirectly). Most of the genes involved in signaling were induced 24 h after the *Bs* MBI600 application. Transcription factors with ethylene response (Csa_2G138780, Csa_7G047400) and LRR proteins (Csa_3G115090, Csa_7G452180) showed a higher expression in cucumber roots. Genes putatively encoding proteins, such as RING-H2 finger (Csa_2G301540) and Jasmonate-induced (Csa_1G642550), were found to be up-regulated 48 h post-application (Table 1). Up-regulated genes related to plant growth were separated into two groups. The first group included genes involved in nutrient up-take, such as potassium channel SKOR (Csa_5G409690), potassium transporter 5 (Csa_4G007060), and zinc finger protein GIS4 (Csa_5G609820). In the second group, various genes involved in plant hormone production were observed to be up-regulated at both time points, such as indole-3-acetic acid-induced protein ARG7 (Csa_7G007930) and auxin-responsive proteins (Csa_2G011420, Csa_3G035310). Genes encoding proteins involved in defense mechanisms, such as peroxidase (Csa_2G406640), endo-1, 3(4)-beta-glucanase (Csa_5G643380), pathogenesis-related protein PR-4 (Csa_2G010370) and thaumatin-like (Csa_3G743950), were also identified (Table 1).

### 2.4. Gene Ontology Enrichment Analysis of DEGs

A gene ontology (GO) enrichment analysis was performed across the treatments at two time points compared to the untreated control plants. The up- and down-regulated genes, classified according to biological process and functional category, are presented in Appendix A. Genes related to the regulation of biological processes, nitrogen compound metabolic processes, and cellular processes were the most enriched at 24 and 48 h post-application, DEGs. According to these functional categories, up-regulated DEGs were enriched only at 48 h post-application in four categories: transcription regulator activity, DNA binding, DNA-binding transcription factor activity, and nucleic acid binding (Appendix A). At 24 h post-application, the down-regulated DEGs were enriched in various categories, such as the cellular macromolecule metabolic process, the dominant in biological process, and organic cyclic compound binding in functionality. On the contrary, at 48 h post-application, there were fewer genes enriched in categories, with the majority of them involved in biosynthetic processes (Appendix A).

### 2.5. KEGG Pathway Analysis

To determine the role of highly expressed DEGs in metabolic pathways, a KEGG analysis was performed. Eight ethylene-related genes were induced significantly 48 h after the application of *Bs* MBI600 (Csa_4G641590, Csa_6G491020, Csa_3G389850, Csa_2G138780, Csa_2G354000, Csa_7G047400, Csa_5G167120, Csa_1G597730). According to the KEGG pathway, the genes were associated with plant immunity and more specifically with MAPK signaling (Figure 4). Regarding DEGs associated with signaling, two genes were found to be up-regulated at time point 24 h and induced receptor-like serine/threonine-proteins kinases (Csa_6G344310, Csa_3G115090). KEGG analysis enriched those genes in plant-pathogen interactions and localized in the bacterial-induced pathway. Finally, five DEGs expressed at both time points were related to auxin-induction (Csa_2G257100, Csa_2G011420, Csa_3G035310, Csa_3G866530, Csa_3G883020) and enriched in the plant hormone signal transduction pathway (Figure 4).

### 2.6. Gene Validation with qRT-PCR

To validate the accuracy and reproducibility of the RNA-seq results, we selected eight genes related to defense and plant growth mechanisms for qRT-PCR assays. For all four defense-related genes tested (gluA, PR4, PO and thaumatin), the transcript levels were found to be increased even 24 h post-application. However, in all but the thaumatin defense genes, the higher induction rate was observed at 48 h post-application (Figure 5). Most of the auxin-related genes were induced at the early time point, with the exception of the auxin-responsive SAUR, which showed higher transcript levels at 48 h post-application (Figure 5). These results were consistent with the data obtained from the RNA-seq analysis.

### 2.7. Metabolite Profile in Cucumber Plants after Application of Bs MBI600

A polar metabolite profile was determined in the cucumber leaves that were simultaneously sampled for the construction of both a metabolite and transcript profile, 24 and 48 hpa of *Bs* MBI600. GC-MS analysis demonstrated that the application of the microorganism affected the metabolic profile at both timepoints. Forty-four (44) and fifty-six (56) metabolites were respectively identified and organized in five groups based on their chemical nature and structure (water-soluble sugars, sugar alcohols, organic substances, organic acids, and amino acids). More specifically, at the first sampling time (24 hpa), within the water-soluble sugar group, fructose, sorbose, glucose, mannose, talose, and especially allose showed an increase in their amounts compared to the untreated control (Figure 6). Regarding sugar alcohols, an increase in myo inositol was also detected, whereas mannitol and glycerol amounts decreased. Furthermore, within the organic acids group, malic, threonic, erythronic propanoic (syn glyceric), and gluconic acids were present in higher amounts in the cucumber leaves after application of *Bs* MBI600. Interestingly, in the 48 hpi group, the metabolic profile differed importantly. In detail, seedlings treated with the microorganism demonstrated a production of galactose and xylopyranose, whereas these were not detected in the control plant tissues and all other amounts of sugars significantly decreased. The abundance of all amino acids was increased, particularly glutamic and aspartic acids, alanine, isoleucine, valine, citruline, glycine, threonine, leucine, and GABA (Figure 6, Appendix A). In regards to organic acids, there was an important increase in malic, erythronic, propanoic, and butanoic acids.

## 3. Discussion

In this study we investigated the transcriptional responses in cucumber roots induced by the biocontrol agent *Bs* MBI600. It is a well-studied BCA known for its ability to control foliar and soilborne pathogens and to promote the plant growth of vegetable crops, such as tomato [20,22,23]. Our results reinforced and confirmed that *Bs* MBI600 is able to promote plant growth and to enhance the photosynthetic efficiency of cucumber plants. Previous studies have shown that applications of *B. cereus* K46 and *Bacillus* spp. M9 increased photosynthesis rate by 20% in pepper plants [27]. The increase in photosynthesis is frequently associated with a higher N_2_ fixation rate [28]. PGPR strains are reported to fix atmospheric N_2_ in soil and avail it to plants, leading to plant height promotion and an increase in fruiting capacity [29].

Phytohormones play a crucial role in plant growth and development during plant responses to different environmental conditions. PGPR applications can modify plants’ phytohormonal levels under environmental stresses [30]. PGPR applications enhance plant IAA concentrations as a part of the plant growth promotion mechanism [5]. Some PGPR strains supply IAA directly to the host, while others trigger the plant auxin pathways by regulating the expression of auxin-responsive genes [31]. The AUX/IAA gene family represents early auxin response genes encoding nuclear proteins, acting as transcriptional repressors of auxin-responsive genes [32]. Our study identified genes (arg7, saur32, IAA4) belonging to those families and, according to the KEGG analysis, related to tryptophane metabolism. A previous study showed that inoculation of wheat plants with *B. subtilis* LDR2 enhanced IAA content and reduced ABA/ACC content, the modulating expression triggered by regulatory component (CTR1) of ethylene signaling pathway and DREB2 transcription factor [33]. Similarly, on rice seedlings, higher expression levels of the *OsIAA1* gene were observed 10 days after the plants’ treatment with *B. altitudinis* FD48 [34]. *OsIAA1* is a member of the Aux/IAA family and induced by phytohormones, such as IAA [35].

In regards to metabolomic profile, the cucumber plants inoculated with *Bs* MBI600 24 hpi exhibited a higher abundance of the sugars fructose, glucose, and mannose, as well as the sugar alcohol myo-inositol. These metabolites participate in crucial pathways, such as carbon metabolism and cell wall component biosynthesis [36]. Indeed, fructose and glucose function as positive signals, regulating growth parameters and metabolic responses [36,37,38], while fructose, mannose, and myo-inositol are precursors of cell wall components [35,39]. These changes denoted the stimulation of dynamic processes, such as cell wall remodeling and/or cell proliferation in response to inoculation with *Bs* MBI600. These findings are probably further connected to the induction of auxin-related genes detected in the same treatment, and supported by the increase in net photosynthesis in the inoculated plants, as mentioned above. It is worth mentioning that cell wall reconstruction can be stimulated by plant-microbe interactions, particularly during colonization of plant tissues by microbes [40]. Interestingly, at the second sampling time (48 hpi), the metabolic profile changed. At this time point, the amounts of sugars decreased in the plants colonized by *Bs* MBI600 as compared to control, but the majority of amino acids were over-accumulated in the same samples, suggesting a prominent change in amino acid metabolism. Among them, the accumulation of glutamate and aspartate, both involved in nitrogen assimilation, indicates plant growth enhancement after PGPR inoculation. On the other side, GABA and proline increased amounts while both acting as priming stimuli [41,42], indicating that the PGPR strain not only enhanced growth features in the plants, but also contributed to its priming, which in turn was a part of ISR induced by the PGPR strain.

PGPR have the ability to increase the availability of nutrients localized in the rhizosphere [43]. The genome analysis of the *Bs* MBI600 strain identified various genes that are involved in potassium, magnesium, nitrate, and phosphorus uptake [18]. Transcriptοme analysis in cucumber roots revealed the induction of potassium transporter genes in *Bs* MBI600-treated plants. Previous studies proved that bacteria belonging to *Bacillus* spp. could regulate high-affinity potassium transporter 1 (HTK1), which modulates Na^+^/K^+^ homeostasis, to mitigate drought stress [44,45]. Potassium plays a key role in stomatal opening and osmotic balance, and controls the transpiration rate in plants under drought stress [46]. Up-regulated genes involved in potassium transport may play a role in the signal movement through the plant vascular system. Induced systemic resistance (ISR) utilizes organic acids and plant hormones (salicylic acid, jasmonic acid, and ethylene) to stimulate host plant defense responses against a variety of plant pathogens [47,48,49]. In previous studies, it has been reported that various Gram-negative *Pseudomonas* spp. and Gram-positive PGPR *Bacillus* spp. have the ability to trigger ISR mainly through JA/ET- and/or SA-dependent signaling pathways [14,50,51,52]. In the present study, various genes involved in signaling pathways were induced in *Bs* MBI600-challenged cucumber roots. Among them, genes belonging to ERF family (Ethylene transcription factors) were included, RING-H^2^ finger proteins, receptor-like serine threonine kinases (LRR) and jasmonate-induced proteins. ERFs are known to act either as activators or repressors of plant defense responses against the biotic stresses caused by fungi, bacteria, and viruses [53]. In addition, ERF proteins (ERF1 and ERF2) activate plant defensins, such as PDF1.2, whereas other *ERF*s (ERF3 and ERF4) are known to repress gene expression and plant defense systems [54]. This is the first report of the induction of ERF proteins after applying a BCA on cucumber plants, confirming a previous report related to the induction of the same genes by *Bs* MBI600 treatments on tomato plants [21]. The induction of *ERF* genes on *Bs* MBI600-treated plants could lead to the activation of defense mechanisms, providing cucumber plants a resistance machinery against plant pathogens. Receptor classes, such as nucleotide-binding site leucine-rich repeat (NBS-LRR) receptors and histidine kinase receptors, can mediate responses to organic chemicals, such as the hormones ethylene or cytokinin, and confer resistance to pathogens [55,56]. Similar genes were found to be induced following applications of *Bacillus* spp. For instance, *Oryza sativa* plants treated with the PGPR strain *B. amyloliquefaciens* FZB24 showed an up-regulated expression pattern in two LRR receptor-like serine/threonine protein kinase genes [57]. Similarly, the *WRR4* gene, that encodes a TIR-NB-LRR protein, was induced in *Arabidopsis* plants, following inoculation with *B. megaterium* BP17 [58]. It is also worth mentioning that isoleucine levels (Ile) increased at 48 hpi, which may indicate JA signaling, since conjugates of the two molecules under the catalysis by jasmonic acid-amino synthetase JAR1 to form JA-Ile, a bioactive molecule of jasmonates (JAs) that has been previously reported [59].

Our study identified Elongation Factor Receptors (EFR), which, according to KEGG analysis, were involved in the plant-pathogen interaction pathway. Interestingly, they were related to bacterial elongation factor (EF-Tu) effectors, including important virulence traits in Gram-positive and Gram-negative pathogenic bacteria [60]. Moreover, in our study, many RING proteins were identified as induced genes in *Bs* MBI600-treated cucumber roots. RING proteins regulate disease resistance in plants by mediating proteolysis of the negative regulator VpWRKY11 through degradation by 26S proteasome [61]. In addition, RING-H2 finger E3 ubiquitin ligase, OsRFPH2-10, is involved in antiviral defense at early stages of rice dwarf viral infection [62]. These resistance mechanisms may offer to *Bs* MBI600-treated plants a resistance that triggers against a wide range of pathogens, including both biotrophs and necrotrophs. KEGG analysis in this study identified genes that are localized in MAPKs pathways. Plant MAPKs are usually localized to the cytosol and/or nucleus, and in some instances, they may also translocate from the cytosol to the nucleus [63]. Upon detecting environmental changes at the cell surface, MAPKs participate in the signal transduction to the nucleus, allowing adequate transcriptional reprogramming. An increasing body of evidence suggests that a subset of plant responses to biotic and abiotic stresses is shared, such as the generation of reactive oxygen species (ROS) and the activation of early defense genes. MAPKs are likely to be one of the converging points in the defense-signaling network [64].

Moreover, the increased levels of threonine after *Bs* MBI600 inoculation may also suggest the activation of ROS-triggered signaling pathways through serine-threonine protein phosphatases metabolism [65]. Evidence of defense pathways activation is also supported by the presence of octadecatrienoic acid after *Bs* MBI600’s inoculation [66]. Previous studies have reported that MAPKs could be activated by external sensors to cellular responses [67]. Otherwise, some studies verified that MAPKs were involved in the interaction between plant and pathogens, and they played key roles in plant response to pathogen invasion. For instance, many effectors secreted via pathogens have been found to inhibit MAP kinase cascade. In addition, MAPKs are involved in the interaction between plants and pathogens, playing a key role in pathogenicity [68]. This is only the second report of MAPKs’ activation with *Bacillus* species applications. In a previous study, applications of *B. velezensis* F21 on watermelon plants led to an up-regulation of various DEGs related to the MAPK pathway [24]. Furthermore, in our study, various plant defense genes were found to be over-expressed, such as peroxidase (POD), β-1,3-glucanase, thaumatin-like, and pathogenesis-related (PR4). The expression of these proteins is related to the defensive responses of plants against fungal infections [69,70]. Such findings are in consistence with the findings of previous reports that suggest plant treatments with several *Bacillus* strains, such as *B. pumilus* SE34 or *B. subtilis* SG_JW.03, induced the accumulation of β-1,3-glucans or elevated expression levels for *PR-1* and *PR-4* [71,72]. The accumulation of D-allose, 24 hpi only in plants inoculated with the PGPR strain constitutes another piece of evidence showing the importance of *Bs* MBI600 as a biocontrol agent, since it has been found that this rare sugar molecule triggers plant defenses against fungi and bacteria, by inducing the production of reactive oxygen species, lesion mimic formation, and PR-protein gene expression [73,74].

## 4. Materials and Methods

### 4.1. Plant Material and PGPR Application

For the transcriptomic analysis of roots, cucumber plants (*Cucumis sativus* L.) cv. Bamboo (Syngenta) at the first leaf stage were used. Bacterial cultures were prepared in flasks containing Luria Broth (LB) medium and shacked overnight at 37 °C. The suspension was then centrifuged at 4000× *g* for 5 min and the pellet was re-suspended in dd H_2_0, until the OD (measured at 600 nm) of the culture reached values of 0.8 (10^8^ cfu mL^−1^). Ten mL of the bacterial suspension were applied in each pot by soil drenching. Seedlings were kept under greenhouse conditions between 20 and 25 °C with a 16/8 h photoperiod cycle and 60–70% RH.

### 4.2. Plant Growth and Photosynthesis Parameters

The effect of *Bs* MBI600 on cucumber plants growth was assessed by measuring the following growth and photosynthesis parameters: shoot height, root length, shoot fresh weight, root fresh weight, net photosynthesis, and chlorophyll content index (CCI). Cucumber seeds were individually sown in plastic pots containing 80 cm^3^ of a 5:1 mixture of peat and perlite. Bacterial cultures were prepared as described above and ten ml of the bacterial suspension were applied in each pot, just after sowing, by soil drenching. The application was repeated 20 days after sowing. In addition to *Bs* MBI600, the commercially available *Bacillus amyloliquefaciens* QST713 (thereafter *Ba* QST713) strain (Serenade ASO, 1.34SC, BAYER Crop Science, Germany), was included in the experimental design as a reference biological treatment. The plants were grown under glasshouse conditions for 35 days and then the parameters regarding growth and photosynthesis were assessed. Measurements of cucumber growth characteristics were conducted as described previously [18]. The net photosynthetic rate (Anet, mmol m^−2^ s^−1^), was measured, using LI-6200 (LICOR, Lincoln, NE) under the following conditions: T leaf = 29 °C, RH = 70%, light = 1300 lux. CCI was measured, using a CCM-200 chlorophyll meter (Optiscience, USA). Both measurements were taken from the third true and fully expanded leaf. Ten replicate plants were used per treatment and the experiment was repeated twice.

### 4.3. Metabolite Extraction, Derivatization and Profiling after Gas Chromatography–Mass Spectrometry (GC–MS) Analysis

Six seedlings per treatment were immediately homogenized under liquid nitrogen subdivided into two technical subsamples (0.5 g each) and stored at −80 °C for polar metabolite analysis. This was performed by GC-MS analysis after derivastization as previously described by Ainalidou et al. (2016) [75]. In brief, frozen samples were transferred into 2-mL screw cap tubes with 1400 μL of 100% methanol (−20 °C), and adonitol (100 μL of 0.2 mg mL^−1^ aqueous solution) was added as internal quantitative standard. Samples were then incubated at 70 °C for 10 min and centrifuged at 11,000× *g* (4 °C) for 10 min. Supernatants were transferred to glass vials with 750 μL chloroform (−20 °C) and 1500 μL distilled H_2_O (4 °C), and after centrifugation at 2200× *g* (4 °C) for 15 min, 150 μL from the upper phase were transferred into new glass vials. After overnight drying in a vacuum desiccator, derivatization was performed with 40 μL of 20 mg mL^−1^ methoxyamine hydrochloride (Sigma Aldrich, St. Louis, MI, USA) in pyridine, incubated in a water bath (37 °C, 2 h). For the completion of derivatization, samples were treated with 70 μL of N-Methyl-N-trimethylsilyl-trifluoroacetamide (MSTFA reagent; Supelco Bellefonte, PA, USA), (37 °C, 30 min). The aliquots were then transferred into 1.5 mL autosampler vials with glass inserts and subjected to GC–MS analysis.

Chromatographic separation and identification of compounds was performed on a Trace GC Ultra-Gas Chromatograph (Thermo Finnigan, San Jose, CA, USA) coupled with a Trace ISQ mass spectrometry detector, a TriPlus RSH autosampler, and an Xcalibur MS platform. One-μL samples were injected with a split ratio of 70:1. Separations were carried out on a TR-5MS capillary column (30 m × 0.25 mm × 0.25 μm, film thickness 0.25 μm). Temperature of injector was 220 °C, of ion source 230 °C, and of interface 250 ^o^ C. Helium was used as a carrier gas at a constant flow rate of 1 mL min^−1^. The GC oven temperature was programmed as follows: initial temperature was 70 °C and held for 5 min, then increased to final temperature 240 °C at a rate of 8 °C min^−1^, and held at 240 °C for 15 min. After 5 min solvent delay, mass range of *m*/*z* 50–600 was recorded. The peak area integration and chromatogram visualization were performed using Xcalibur processing program. Quantification of the detected metabolites was based on comparisons with adonitol and expressed as relative abundances. For peak identification and mass spectra tick evaluation, the NIST11 database (National Institute of Standards and Technology, Gaithersburg, MD, USA) was used. Mass spectra were cross-referenced with those of authentic standards in the Golm metabolome database (gmd.mpimp-golm.mpg.de/2021) [76,77].

### 4.4. RNA Extraction, Library Construction and RNA Sequencing

The roots of cucumber plants were collected for RNA sequencing 0, 24, and 48 h after the drenching application of *Bs* MBI600. The collected roots were soaked in liquid nitrogen. Total RNA was extracted using the Trizol method according to manufacturer` instructions (TRItidy G™, Germany), following a modified protocol with Monarch Total RNA Miniprep Kit (NEB #T2010) to increase the quality of the extracted RNA. The quality of RNA was assessed on 1% agarose gels. Before the submission for sequencing, the quantity and quality of RNA were verified using an RNA Nano 6000 Assay Kit of the Bioanalyzer 2010 system (Agilent Technologies, CA, USA). NGS libraries were generated from 500 ng input total RNA according to manufacturer protocol in the QuantSeq 3′ mRNA-Seq Library Prep Kit FWD for Illumina kit from Lexogen. Libraries were run on Illumina 500 on 1 × 75 High Flowcell. RNA sequencing service was provided by the Institute of Molecular Biology and Biotechnology Foundation for Research and Technology (Heraklion, Greece) in IMBB Genomics Facility.

### 4.5. Gene Ontology and KEGG Enrichment Analysis of DEGs

Gene ontology (GO) enrichment analysis of differentially expressed genes was implemented by ShinyGo app [77]. GO terms with a corrected *p*-value less than 0.05 were considered significantly enriched by differential expressed genes. KEGG (Kyoto Encyclopedia of Genes and Genomes) is a database resource for understanding high-level functions and utilities of biological systems, such as the cell, the organism, and the ecosystem, from molecular-level information, especially large-scale molecular datasets generated by genome sequencing and other high-throughput experimental technologies (http://www.genome.jp/kegg/ Release 99.1, 2021). KOBAS software was used to test the statistical enrichment of differential expression genes in KEGG pathways [6].

### 4.6. qRT-PCR Assays

The RNA-seq data were validated with quantitative Real Time PCR (qRT-PCR). Eight genes were selected for qRT-PCR among those that showed an up-regulation pattern following *Bs* MBI600 treatment. All primers used for qRT-PCR are listed in Appendix A. The qRT-PCR reactions were carried out using a StepOne Plus Real-Time PCR System (Applied Biosystems, USA) using a SYBR green-based kit (Luna^®^ Universal One-Step RT-qPCR Kit, NEB, UK) according to the manufacturer’s instructions. Amplification conditions were 55 °C for 10 min, 95 °C for 60 s, followed by 40 cycles of 95 °C for 5 s and 60 °C for 30 s. In all the reactions, samples were run in triplicate. The threshold cycle (CT) was determined using the default threshold settings. The 2^−ΔΔCt^ method was employed to calculate the relative gene expression levels [26]. Cytochrome oxidase (cox) gene was used as endogenous control.

### 4.7. Data Analysis

Bioinformatic analysis was conducted as previously described [19]. Fastqc was used for quality assessment of sequenced reads. Processed reads from Illumina-BaseSpace were quality assessed using Fastqc (https://github.com/s-andrews/FastQC) and mapped to Cucumis_sativus genome (ensemble, release-47. Cucumis_sativus. ASM407v2) using hisat2 version 2.1.0 (--score-min L 0, −0.5) [78,79]. Gene counts were computed with htseq-count (-s yes, version 0.11.2) [80]. Differential analysis was performed with edgeR in SARTools [81].

Data of the independent replications on plant growth and photosynthetic parameters were combined after testing for homogeneity of variance using Levene’s test. The combined data were then subjected to one-way analysis of variance (ANOVA). Fisher`s LSD test at *p* = 0.05 was used for comparison of means. Data from the GC–MS based metabolite profiling were analyzed by analysis of variance (ANOVA), and mean values were computed from six replicates. The differences between treatments’ mean values were compared using *t*-test. The significance level in all hypothesis testing procedures was predetermined at *p* ≤ 0.05. All statistical analyses were performed with the SPSS v 25.0 software (SPSS Inc., Chicago, IL, USA).

## 5. Conclusions

In this study, we investigated the transcriptome changes caused by the recently introduced to the market BCA *Bs* MBI600 on cucumber plants. We found that the BCA was able to induce genes involved in plant growth and the defense of plants, providing insights into the molecular mechanisms of the interaction between the host and the BCA. On the metabolic level, PGPR treatment seems to induce an extensive reprogramming involving hydrocarbon accumulation 24 hpi and consequentially an increase in amino acids 48 hpa. Thus, the results of the two “omic” analyses indicated that *Bs* MBI600 colonization triggers plant responses in two directions, the induction of growth indices, and in parallel, an increase in the alertness of the same plants, which are expected to be more robust and tolerant under stress conditions. Such findings provide further evidence that this biocontrol agent could be compatible with sustainable cucumber cultivation.

## Figures and Tables

**Figure 1 plants-11-01218-f001:**
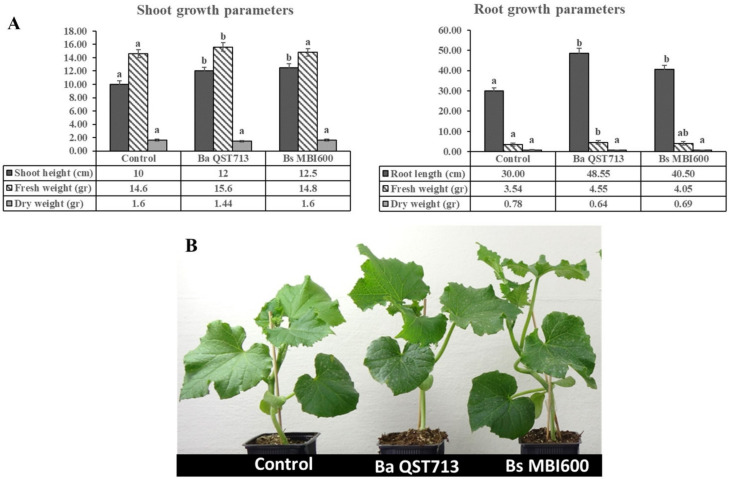
Effect of *Bacillus subtilis* MBI600 (*Bs* MBI600) applications on cucumber plants. (**A**) Shoot and root growth parameters as compared to the growth of untreated control plants and *Bacillus* amyloliquefaciens QST713 (*Ba* QST713)-treated plants. Different letters on the columns indicate significant differences according to Fisher`s LSD test (*p* < 0.05). Vertical lines indicate the standard error of the mean. **(B**) Representative plants treated with *Bs* MBI600 and *Ba* QST713 as compared to untreated plants 35 days after sowing.

**Figure 2 plants-11-01218-f002:**
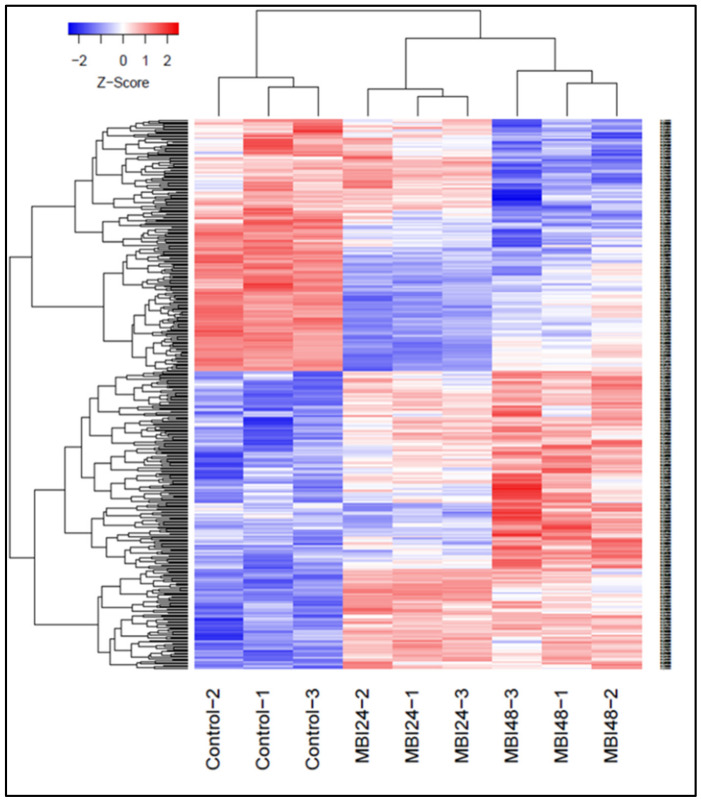
Genes that displayed a greater than two-fold difference (*p* < 0.05) in expression were identified and plotted in a heat map, and the gene expression in conditions and time points was compared. Red and blue colors represent genes with up expression or down expression values, respectively.

**Figure 3 plants-11-01218-f003:**
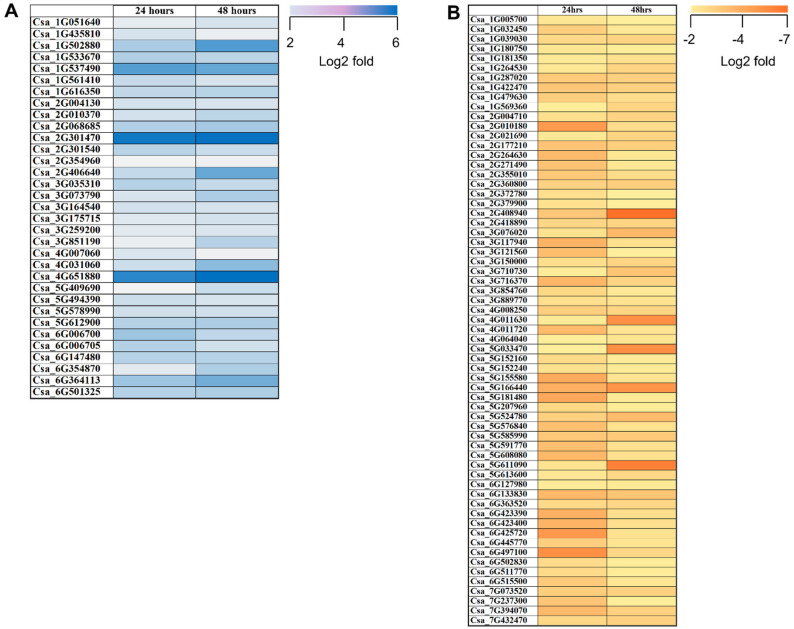
Heat map showing transcription profiles of cucumber roots treated with the biocontrol agent *Bacillus subtilis* MBI600 at two time points (24 and 48 h post-application); (**A**) up-regulated and (**B**) down-regulated genes. Data were normalized with cucumber roots at 0 h post-application (adjusted *p*-value < 0.05, absolute log2 fold change > 2). Blue and yellow colors represent up- or down-regulated genes, respectively.

**Figure 4 plants-11-01218-f004:**
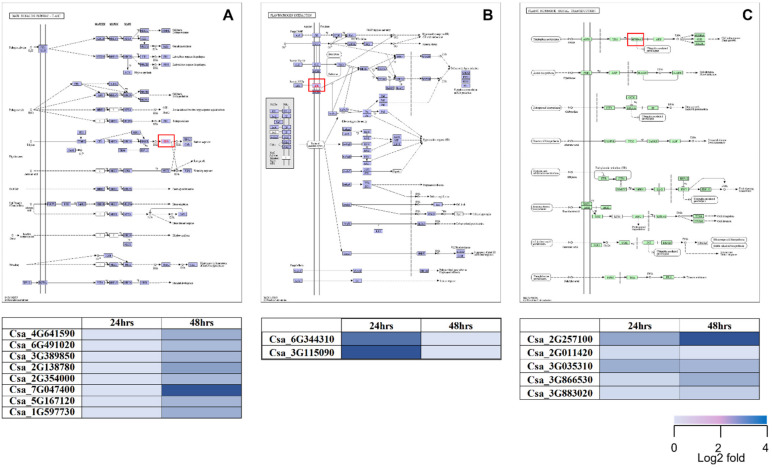
KEGG analysis of DEGs related to (**A**) MAPK signaling, (**B**) plant-pathogen interaction and (**C**) plant hormone signal transduction pathways. Heat maps show the expression level of DEGs involved in each KEGG pathway.

**Figure 5 plants-11-01218-f005:**
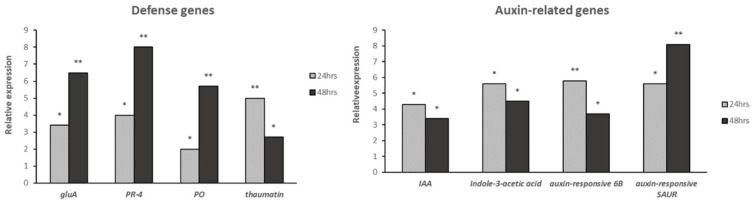
Expression levels of auxin and defense proteins encoding genes in the roots of cucumber plants treated with *Bacillus subtilis* MBI600, 24 and 48 h post-application. Gene expression levels were normalized by respective expression before the BCA application (time point 0 h). The cDNA samples were normalized using the endogenous cox gene and the expression levels were calculated using the 2^−ΔΔCt^ method [26]. Asterisks on the columns indicate statistically significant differences according to Tukey’s test (*p* < 0.05).

**Figure 6 plants-11-01218-f006:**
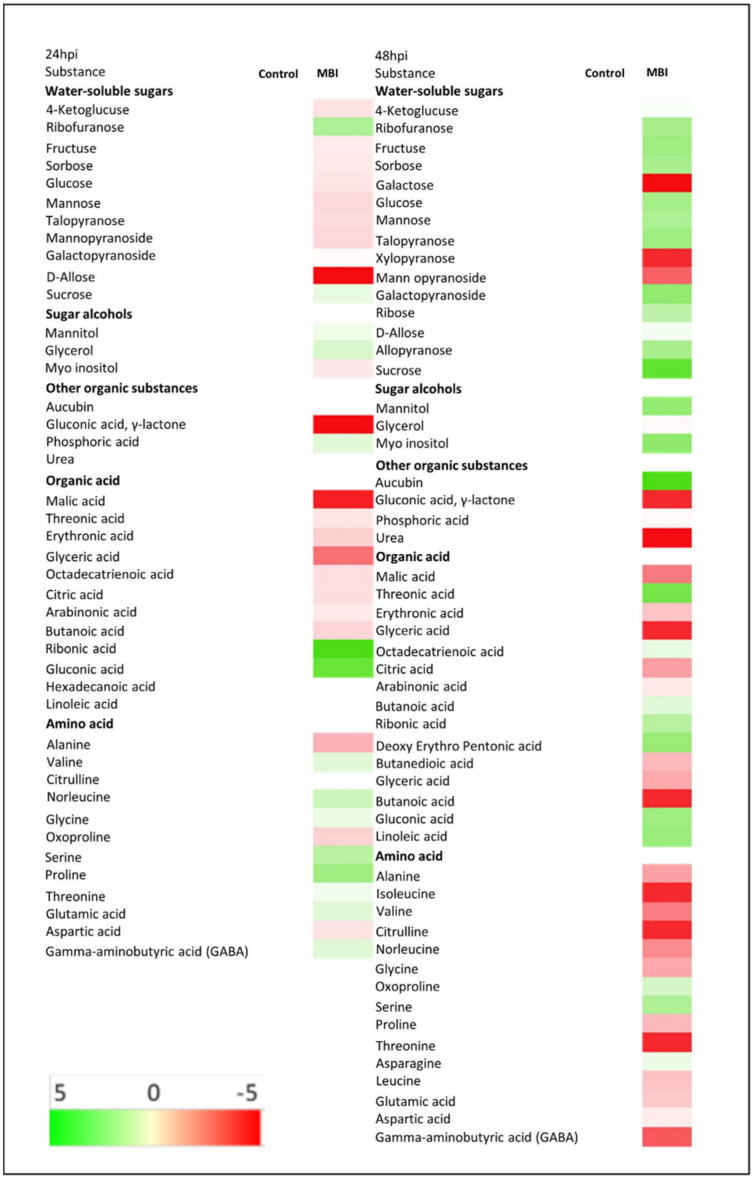
The heatmap of metabolites changes in cucumber leaves of plants treated with *Bacillus subtilis* MBI600 compared to non-treated plants, sampled, and analyzed by GC-MS, 24 (24 hpi) and 48 (48 hpi) hours post-application of the biocontrol agent. A decrease in metabolites concentration is indicated by a green color and an increase is indicated by a red color, as explained by the color scale at the bottom of the figure.

**Table 1 plants-11-01218-t001:** Top up-regulated genes encoding proteins involved in signaling, plant growth, and defense in cucumber roots treated with *Bacillus subtilis* MBI600.

Gene ID	Protein	Function	Expression Pattern ^a^
Csa_2G138780	ethylene-responsive transcription factor ERF062	Signalling	24
Csa_3G115090	L-type lectin-domain containing receptor kinase IX.1		24
Csa_7G452180	LRR receptor-like serine/threonine-protein kinase IOS1		24
Csa_7G047400	ethylene-responsive transcription factor ERF054		48
Csa_1G642550	jasmonate-induced protein		48
Csa_2G301540	RING-H2 finger protein ATL78		24 + 48
Csa_3G175715	pentatricopeptide repeat-containing protein		24
Csa_7G007930	indole-3-acetic acid-induced protein ARG7	Plant growth-related mechanisms	24 + 48
Csa_5G409690	potassium channel SKOR		24 + 48
Csa_2G011420	auxin-responsive protein IAA4		24 +48
Csa_4G007060	potassium transporter 5		24 + 48
Csa_3G035310	auxin-responsive protein SAUR32		24 + 48
Csa_3G866530	auxin-responsive 6B		24 + 48
Csa_5G609820	zinc finger protein GIS4		48
Csa_2G406640	peroxidase 55	Defense-related mechanisms	24 + 48
Csa_3G743950	thaumatin-like protein 1b		24
Csa_7G044780	peroxidase 28		48
Csa_5G643380	endo-1,3(4)-beta-glucanase 3		48
Csa_2G010370	pathogenesis-related protein PR-4		24 + 48

^a^ The expression pattern indicated the time post-inoculation with *Bs* MBI600, when the specific genes were up-regulated.

## Data Availability

The datasets supporting the results of this article are available in the NCBI (BioProject ID:PRJNA765367, BioSample accession: SAMN21555696).

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
