# Peer review of "Root Transcriptional and Metabolic Dynamics Induced by the Plant Growth Promoting Rhizobacterium (PGPR) Bacillus subtilis Mbi600 on Cucumber Plants"

_plants, 2022, doi:10.3390/plants11091218_

Round 1

Reviewer 1 Report

The current study represents the PGP mechanisms of a PGPR strain with combined transcriptional and metabolic analysis. There are many similar research works,and the story is also similar. However, this article was well organised, and provided some interesting insight to the mechanisms of PGPR. Therefore, I chose  a positive opinion. The only thing to explain is why chose GC-MS but not MS -MS to perform metabolic analysis . MS-MS should be more suitable.

Author Response

Dear reviewer thank you for your comments.

As it for your comment, we could say that GC-MS is a well established method for identifying plant metabolites and our lab is equipped for this type of analysis. 

Reviewer 2 Report

Abstract:
-Consider focusing the results on the cucumber research field, because if not, there is not any novelty in this work (this strain and its effects are very well characterized in other plant species).

Introduction:
General: The order and relevance of the topics seems not very clear. It fells like a random placement of interesting concepts, I think this requires a little reformulation. Last 2 paragraphs are great, keep there.
-Line 49: Not a 'theory', has been largely demonstrated. 
-Line 53: I understand you need to justify the relevance of this work, but the knowledge is not that limited. This could cause overrating on your work expectations. Please, moderate this kind of assessments.
-Line 55-61: This part is too wide and touch random topics. Not only Gram-negative can do what you claim, but you don't mention Gram+, despite your model strain is one. Then you jump to a brief description of other phytohormones, that is a little out of place (maybe at the beginning and then focus on IAA?). Moreover, IAA is relevant to concede space here, but also many other regulation pathways same relevance in your results, consider to rearrange.
-Line 62-69: Out of place, consider to rearrange earlier than part of this biotechnology description.

Results:
2.1 - This section, despite the relevance, has been previously reported for these strains and for similar ones. BE careful with the assessments in the 'Discussion' (Lines 106-107 seems like this is the first time, I would use 'reinforce', 'assess' ... in our genotype..., something like this).
Fig. 6 - Color scale is not included

Discussion:
-General: avoid including explanations or data that seems more like 'Introduction'. Here some statements are not even part of the discussion, properly said. It's just extra information (particularly at the beginning of in page 11). This doesn't mean authors cannot include data as prelude, but not as a second introduction.

Materials and Methods:
- Plant material: Not full required, but first time mentioned in materials, it's recommended to use scientific name for cucumber, as well as the source of origin (seed bank, institution...)
- Degrees symbol is not correct. Also, is not separated from numbers (no space).
- Resuspend in dd H20 is not the best practice. I understand that Bacillus strains can handle, but this osmotic shock could not be the best way to proceed or reproduce with other Bacillus strains or other kind of bacteria.
- Should include the meaning of 0.8 OD in CFU to be consistent and comparable to other works.
- Unnecessary to repeat concepts as how to inoculate or fertilization is carried out.
- Consider dividing plant and bacteria material and management. As it is now, it could create confusions.
- For g-forces, consider to use 'x g' instead of just 'g'. Thus, 'g' should be in italic.

Conclusions:
Line 518-519: This is a brief, not a conclusion :) Maybe change the initial 'In conclusion, ...'

Author Response

Dear reviewer thank you for your comments.

"Please see the attachment

Reviewer 3 Report

Recently, researchers have been actively studying the influence of the rhizosphere and endophytic microbiome of plants and individual isolates on the physiological functions of plants. A special place among such works is occupied by studies where the authors pay attention to changes in the transcriptional activity of genes and the content of plant metabolites. In this regard, the work of Samaras A. et al. “Root transcriptional and metabolic dynamics induced by the Plant Growth Promoting Rhizobacterium (PGPR) Bacillus subtilis MBI600 on cucumber plants” is in the trend of time. The authors of the manuscript conducted an analysis of the effect on the physiological parameters of cucumber plants, including transcriptome analysis and changes in the metabolomic profile, after inoculation of Bacillus subtilis MBI600, a commercialized plant growth-promoting bacterial species. The article is written quite clearly, the arguments given by the authors are convincing and in general, the article can be published in the journal after correcting minor comments:

  1. In the introduction section, the authors indicated that the Bacillus subtilis MBI600 strain is a commercialized plant growth-promoting bacterial species and is marketed by BASF. However, in the methodological part, the authors do not provide data on the origin of this strain. I would like to note that readers often carefully study the methodological part of the work.
  2. In the methodes, it is probably necessary to mention the Latin name of the cucumber, as well as the source of origin of the seeds. And also, at the first mention of a bacterial culture, decipher the name of the strain and its origin, which I indicated in the first remark.
  3. Table 1 shows data on Top-up regulated genes encoding proteins involved in signaling, plant growth and defense, cucumber roots treated with Bacillus subtilis MBI600. These data are very interesting, but they would be even more informative if the authors provided numerical data on changes in their transcriptional activity. I think that modern devices allow you to do this.
  4. In Figure 4, the authors attempted to show the involvement of plant genes responsive to plant treatment with Bacillus subtilis MBI600 strain in plant metabolic signaling pathways responsible for MAP kinase signaling, plant–plant interaction, and hormonal signal transduction pathways. Unfortunately, due to the very small scale, it is extremely difficult for the reader to understand this figure. Is it possible to divide this drawing and show it in a more presentable form.

Author Response

Dear reviewer thank you for your comments.
